# Crosstalk Noise of Octagonal TSV Array Arrangement Based on Different Input Signal †

**Ziyu Liu** [1,2,*], **Han Jiang** [3], **Ziyuan Zhu** [3], **Lin Chen** [1], **Qingqing Sun** [1] **and Wei Zhang** [1]

1   School of Microelectronics, Fudan University, Shanghai 200433, China; linchen@fudan.edu.cn (L.C.);
    qqsun@fudan.edu.cn (Q.S.); zhangwei@fudan.edu.cn (W.Z.)
2   Hubei Yangtze Memory Laboratories, Wuhan 430205, China
3   School of Electronic Information Engineering, Southwest University, Chongqing 400715, China;
    jiang_han1122@163.com (H.J.); zyuanzhu@swu.edu.cn (Z.Z.)
*   Correspondence: liuziyu@fudan.edu.cn; Tel.: +86-189-3030-1560
†   This paper is an extended version of paper published in the international conference: IEEE Electron Devices
    Technology & Manufacturing Conference (EDTM), Chengdu, China, 9 April 2021.

**Abstract:** This paper proposes an octagonal layout for enhancing the ability of resisting electromagnetic interference in Through Silicon Via (TSV) array. The influential factors of crosstalk noise between TSVs are investigated, including the TSV pitch, signal and ground TSVs location, and signal types (single-end and differential signal) by using a coplanar wave guide (CPW) testing structure. These results, based on traditional TSV arrays, show that a staggered TSV layout with differential signals had lower crosstalk noise. On this basis, the octagonal layout of TSV array is proposed and we show that it has obvious superiority in reducing occupied silicon area and crosstalk noise. Compared with traditional TSV arrays, the crosstalk noise is almost reduced by 44%. In order to further reduce the silicon area occupied by TSV without worsening crosstalk noise, the new division TSV structure is proposed in which a large TSV was substituted by four smaller TSVs. The area occupied by a single TSV and TSV array are both reduced by 60% without decreasing signal integrity when the regular TSV in the octagonal layout are replaced by a new TSV structure.

**Keywords:** TSV array; octagonal layout; magnetic interference; crosstalk noise; differential signal; new structure

## 1. Introduction

Three dimensional (3D) integration has become one of the most critical technologies in the post-Moore era. TSV technology occupies an extremely important position in 3D integrated circuit (IC). The vertical interconnection of TSV can reduce the length of the interconnection path and decrease the signal delay. As well, it can save the silicon area and increase integration density. Therefore, TSV has become an important technology in the application of 3D IC [1].

The TSV fabrication process is paid much attention to in recent research, although few studies focus on the signal integrity. However, signal integrity is one of the important criteria to judge TSV electrical performance. The signal is transmitted through the TSV array and thousands of TSVs are placed in the limited silicon chip area. This often causes great electromagnetic interference, including crosstalk noise, which will affect the signal integrity transmitted through TSV [2]. Therefore, optimizing TSV arrays with signal/ground TSVs to reduce the crosstalk noise has large significance in determining signal integrity. However, TSV arrays should have less ground TSV and occupy less silicon wafer area, which conflicts with better crosstalk noise performance. Therefore, lower-crosstalk noise TSV arrays with smaller silicon area have to be investigated deeply [3].

The magnitude of crosstalk noise between TSVs is mainly determined by coupling noise, as shown in Figure 1. The $C_{ox}$, $R_{substrate}$ and $C_{substrate}$ are the capacitive reactance of the dielectric layer, silicon substrate impedance and capacitive reactance, respectively [3].

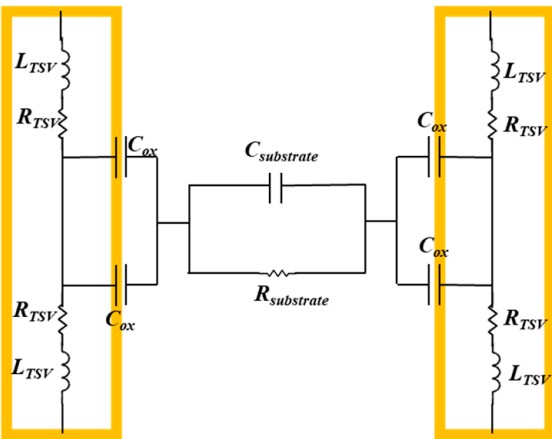

**Figure 1.** Coupling circuit between TSV.

The magnitude of $C_{ox}$ is expressed as [3]:

$$C_{ox} = \frac{2\pi\varepsilon h}{\ln\left[(2t_{ox}+D)/D\right]} \tag{1}$$

where h is the height of TSV, D is the diameter of TSV, $t_{ox}$ is the thickness of dielectric layer and $\varepsilon$ is the dielectric constant of Silicon dioxide, respectively [3]. The magnitude of $C_{ox}$ is constant as the $t_{ox}$, h, and $\varepsilon$ are set at a constant value.

Meanwhile, the magnitude of $C_{substrate}$ and $R_{substrate}$ can be expressed as [3]:

$$C_{substrate} = \frac{\pi\varepsilon_0\varepsilon_{r.si}\times h}{\cos h^{-1}\left(\frac{S_{TSV}}{D}\right)} \tag{2}$$

$$R_{substrate} = \frac{1}{\left(\frac{\sigma_{Si}}{\varepsilon_0\varepsilon_{r.Si}}+\omega\tan\delta_{r.Si}\right)\times C_{Si}} \tag{3}$$

where $S_{TSV}$ is the space between two TSVs, $\sigma_{Si}$ is the relative permittivity of silicon substrate, $\omega$ is the applying angular frequency and $\tan\delta_{r.Si}$ is the dielectric loss tangent. The $C_{substrate}$ and $R_{substrate}$ will be increased as $S_{TSV}$ increases based on Equations (2) and (3), which means that the crosstalk noise between TSVs will be reduced.

As the signal frequency increases, the signal integrity tends to be significantly affected by the transmission path due to the crosstalk noise between two TSVs [3]. The coupling model of the TSV array is built to investigate the crosstalk noise of large TSV arrays, which shows that the low-resistivity silicon substrate can enhance coupling noise between two TSVs [4]. Meanwhile, reference [5] reported the crosstalk noise of TSV arrays by building an analytical model based on the theory of multiconductor transmission lines (MTLs), which is more accurate and quicker than the traditional analytical model. To suppress the crosstalk noise, de-embedding test structures and the CPW structure for measuring the crosstalk noise between two TSVs was proposed in the references [6,7]. Compared with the de-embedded structure, the CPW structure can measure crosstalk noise more accurately. The effect of the TSV pitch on the crosstalk noise of mutual TSVs is also reported as being that the crosstalk noise increases as pitch decreases [8]. Compared with the traditional TSV mesh array topology, the hexagonal TSV array can decrease the crosstalk noise and reduce the silicon area [9].

However, research on reducing crosstalk noise in TSV array is seldom found and more research focuses on the process and thermal stress of TSVs in recent years. Meanwhile, there is little current research on using differential signals to TSV, such as references [10,11]. Therefore, this paper proposes an octagonal TSV array arrangement based on the studies referred to above and the CPW structure is used to measure the crosstalk noise, as shown

in Figure 2. The octagonal TSV array with inputs taken as single signal and differential signal both have lower crosstalk noise than traditional TSV arrangements. Meanwhile, the area occupied by the octagonal array is also the smallest. Furthermore, a method to reduce the area without reducing the ability of suppressing crosstalk noise is also proposed by dividing a large TSV into four smaller TSVs, which also largely saves the silicon area.

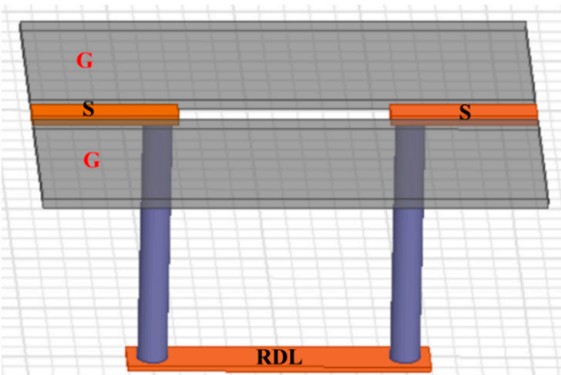

**Figure 2.** Simulation model of CPW structure in HFSS: Ground-Signal-Ground structure.

## 2. TSV Array Design

### 2.1. Octagonal Layout Design

The signal is transmitted through TSV array. Therefore, the TSV array has great impact on the crosstalk noise between TSVs, especially for high TSV density. A reasonable layout of a TSV array can greatly reduce the crosstalk noise between signal TSVs. Therefore, a new octagonal TSV array is proposed in this work and the design is introduced as followed.

To compare the ability of decreasing crosstalk noise among different TSV arrays, the materials and height of TSV are set as shown in Table 1.

**Table 1.** Parameters.

| Name | Materia | Thickness/Height |
|---|---|---|
| Substrate | Silicon | 50 μm |
| Dielectric layer around TSV | Silicon dioxide | 0.5 μm |
| TSV | Copper | 50 μm |
| Dielectric layer on the top | Silicon dioxide | 1 μm |

Traditional layouts of TSV arrays with single-ended signal in an RF circuit are shown in Figure 3a,b, and are treated as the control group. Array 1 has the signals (S)/ground (G) TSVs aligned in parallel with each other (called parallel array) and array 2 has signal (S)/ground (G) TSV staggered with each other (called staggered array), which is used to characterize the crosstalk noise of different TSV arrays. Hexagonal TSV arrays, as used in reference [9], is shown in Figure 3c, which is used to verify the reported result and is also treated as the control group. The new octagonal array of a TSV array is designed as shown in Figure 2d, which is formed by the staggered signal TSV(S) and ground (G) TSV. In the four arrays above, the adjacent TSV remains the same diameter as it is. The pitch of adjacent TSVs keep the same diameter but the pitches between adjacent signal TSVs in arrays 3 and 4 are greater than those of arrays 1 and 2. Due to the larger pitch of the two signal TSVs in arrays 3 and 4, the crosstalk noise is smaller [5]. All these designs are used to access the silicon area occupied by the new octagonal TSV array.

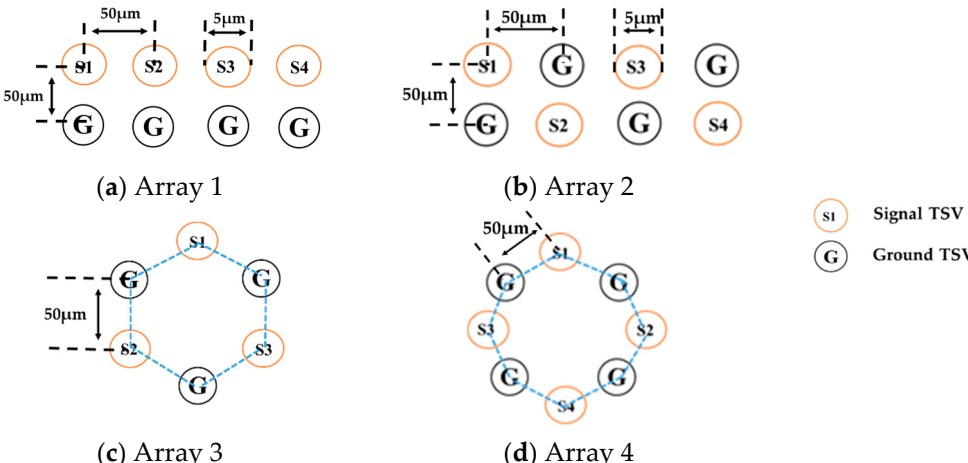

Figure 3. Three TSV arrays: (**a**) S-S layout, (**b**) G-S layout, (**c**) hexagonal layout, (**d**) octagonal layout.

The TSV layout in Figure 4 is used to characterize the function of the differential signal. Figure 4a,b are the units in Figure 3a,b. Figure 4c has the same array as Figure 4b but replaces the single-ended signal with a pair of differential signal because differential signal is reported to decrease the crosstalk noise [10,11].

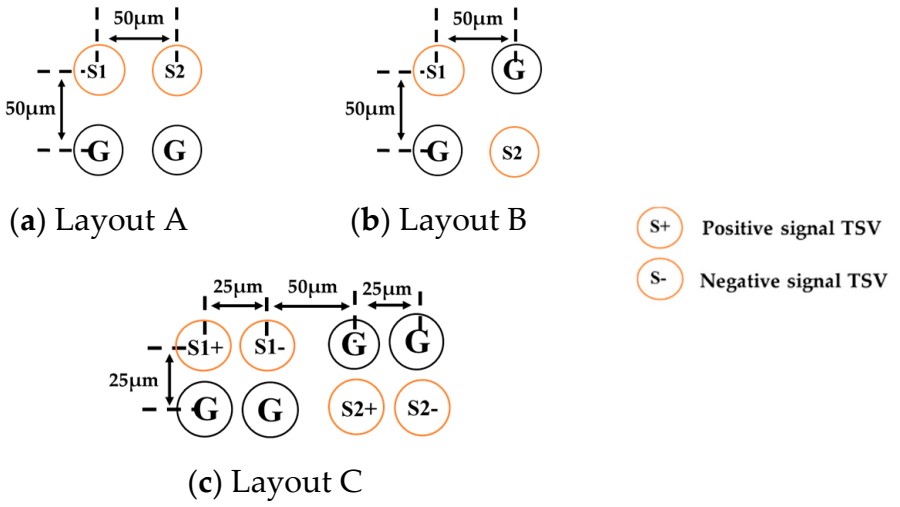

**Figure 4.** Three Layouts of TSV array: (**a**) TSV array side by side, (**b**) Staggered TSV array, (**c**) A pair of differential signal TSVs instead of a single-ended signal TSV staggered array (same silicon area of TSV array).

Figure 5a has the same layout as Figure 4c, which is used to compare crosstalk noise with our octagonal TSV array as shown in Figure 5b. The size design of the TSV array is as follows: the pitch between TSVs for a pair of differential signals and two single-ended signals are 25 μm and 50 μm, respectively. The pitch between signal and ground TSV in a row is 25 μm, and the pitch between ground to ground TSV is 50 μm, as shown in Figure 5. The height and diameter of TSV is 50 μm and 5 μm, respectively, and the oxide thickness is 0.5 μm.

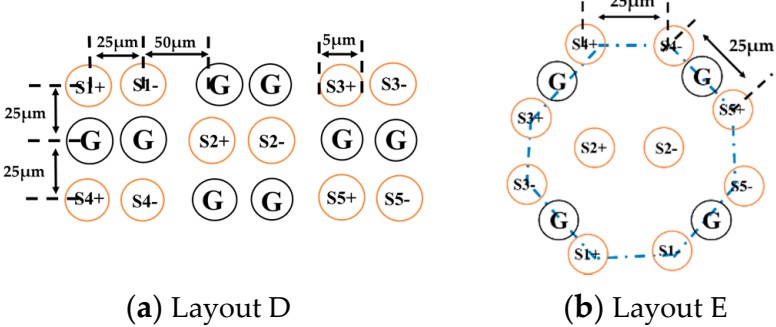

(**a**) Layout D  (**b**) Layout E

**Figure 5.** Two arrays of TSV array: (**a**) five pairs of differential signal TSV in staggered array, (**b**) five pairs of differential signal in octagonal TSV array.

### 2.2. New Divison TSV Structure

In order to better reduce the silicon area occupied by the TSV, the new division structure of the TSV is proposed as shown in Figure 6b, in which a large TSV (diameter is 5 µm) is replaced by four small TSVs (diameters is 2 µm). Compared with the large TSV, the silicon area occupied by the new structure is greatly reduced by 36%, which means that the utilization rate of silicon substrate can be greatly increased. In reference [11], the coupling effect is reduced as the influence path distance becomes larger. Therefore, the influence path from aggressive TSV to victim TSV is designed as shown in Figure 7. The new division TSV structure, replacing the regular TSV structure of layout E, is layout F, as shown in Figure 8. In the existing process, the diameter of the TSV can be manufactured as small as 100 nm by using a Bosch process applied to extremely small CD structures (180 × 250 nm top CD) [12]; thus, the new TSV structure can be manufactured. The smaller TSV pitch is 3 µm and the above TSV pitch reminds the same by 25 µm.

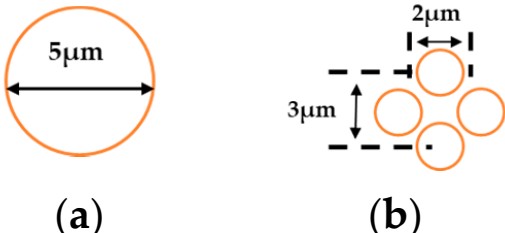

**(a)**  **(b)**

**Figure 6.** TSV structure. (**a**) Regular TSV; (**b**) New division TSV Structure.

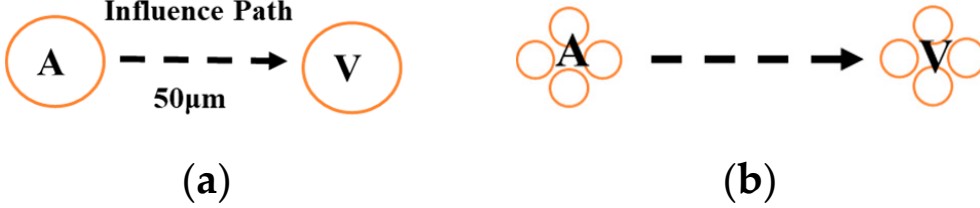

**(a)**  **(b)**

**Figure 7.** The influence path from aggressive TSV to victim TSV: A (Aggressive TSV), V (Victim TSV). (**a**) Two regular TSVs; (**b**) Two division TSVs.

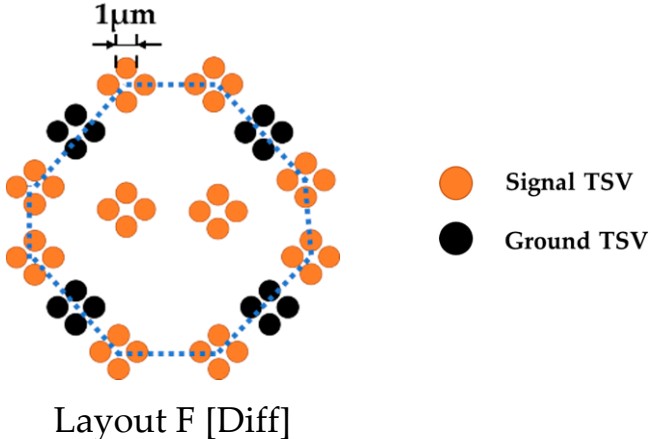

Layout F [Diff]

**Figure 8.** New division structure of octagonal TSV based on differential signals.

## 3. Results and Discussion

The crosstalk noise of traditional TSV arrays (arrays 1 and 2), hexagonal array and octagonal array with single-ended TSV, are shown in Figure 9, obtained by ANSYS HFSS from the American ANSYS Company, Commonwealth of Pennsylvania, US. As the frequency increased, the crosstalk noise of the octagonal array slightly decreased. At low frequencies, the signal was only transmitted in a single TSV, and would not penetrate the silicon substrate to affect other TSVs [3]. Therefore, the crosstalk noise curve of any array overlapped when the diameter was a constant value as shown in Figure 9. Compared with arrays 1 and 2, the crosstalk noise of array 4 was reduced by 15 dB and 5 dB, respectively, when the frequency was over 20 GHz. The main reason for this was that the octagonal structure can reduce the area while increasing the pitch of two signal TSVs, as shown in Figure 3. Compared with array 3 (hexagonal array), the octagonal array could also reduce the crosstalk noise by 2 dB. The main reason for this was that the grounded TSV between two signal TSVs in the octagonal array was closer to the coupling path than the hexagonal TSV; thus, the ability of shielding noise was better even though the space between two signal TSVs was smaller. The crosstalk noise of layout B was higher than layout A by nearly 5 dB when the frequency was over 1 GHz, as shown in Figure 9. The main reasons were as follows: (1) the staggered signal and ground TSV array increased the distance between adjacent signal TSV and signal TSV. (2) The staggered signal and ground TSV array made one signal TSV surrounded by two ground TSVs, which could greatly suppress the crosstalk noise without changing TSV pitch and total array area.

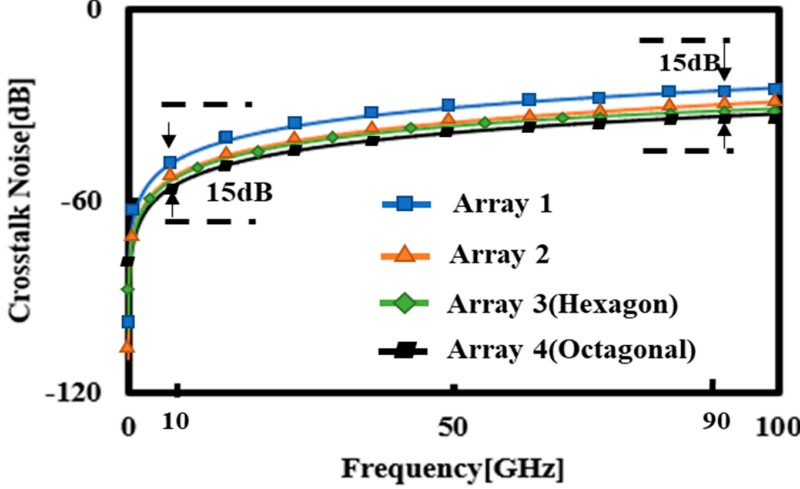

**Figure 9.** The crosstalk noise among arrays 1, 2, 3 and 4.

The crosstalk noise between layout B and layout C(Diff) in Figure 4b,c is shown in Figure 10. When the two single-end TSVs in layout B were substituted by one pair of differential TSVs in layout C(Diff), the crosstalk noise of layout C(Diff) was largely reduced by nearly 15 dB over 10 GHz. The main reason for this was that the differential signal had a great ability of anti-electromagnetic interference, which can effectively decrease crosstalk noise. Notably, the silicon area of layout C(Diff) was equal to layout B.

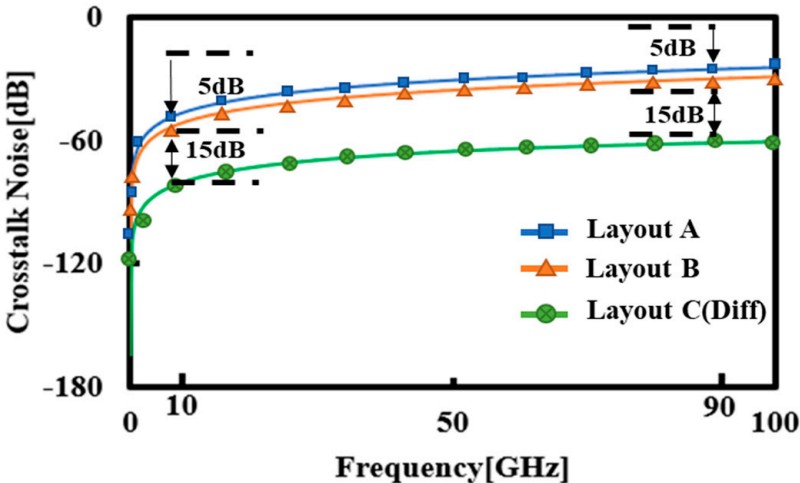

**Figure 10.** The crosstalk noise among layouts A, B and C(Diff).

The results of Figure 9 verified that the staggered TSV array with differential signal had the ability to suppress crosstalk noise, which could be reduced 40% at high frequency. In layout C(Diff), the number of signal TSVs increased but the total area of substrate was the same as for layouts A and B. Therefore, differential staggered arrangement could better suppress crosstalk noise without changing the occupied silicon area. However, the number of grounded TSVs was the same as the signal TSV in the layout C(Diff).

To further save more silicon area by using less ground TSV, differential signals were used in our new octagonal array as shown in Figure 5b. Layout D(Diff) in Figure 5a was a topologic array of layout B, which was to keep the same signal TSV. Five pairs of differential signal TSVs were arranged and the area of the TSV array in layout D(Diff) and E(Diff) was 8750 $\mu m^2$ and 3017.5 $\mu m^2$, respectively. Consequently, layout E(Diff) had a smaller area than layout C(Diff). Besides, it also contained less ground TSV and more signal TSV to transmit data.

Crosstalk noise was further investigated for the layout E(Diff) with less ground TSV and array area. This study chose the representative signal parameters S1,2 and S1,3, where S1,2 meant the signal crosstalk noise between S1 and S2. For the anti-electromagnetic coupling ability of the differential signal, only when the positive and negative signals received the same magnitude of external interference could the internal mutual crosstalk noise be cancelled and, thus, could the differential signal reduce anti-electromagnetic coupling. S1, S2 and S3 were the integration signals of S1+/S1−, S2+/S2 and S3+/S3−. Figure 11a showed the crosstalk noise S2,1 between layout D(Diff) and layout E(Diff). The crosstalk noise of layout E(Diff) was lower than layout D(Diff) with the signal frequency increasing. When the frequency was 10 GHz, the crosstalk noise difference between the two layouts was as high as 45 dB. When the positive and negative signals in a pair of differential signals received the same external interference, the internal crosstalk noises could counteract each other. In the layout E(Diff), S2+ and S2− suffered from the same interference from S1+ and S1−, therefore the crosstalk noises between S2+ and S2− could counteract each other. However, the distance between S1+/S1− and S2+/S2− was inconsistent in layout D(Diff), resulting in a crosstalk noise difference between S1+/S1− and S2+/S2−. Therefore, the crosstalk noise between S2+ and S2− could not be counteracted in the layout D(Diff). Figure 11b showed that the difference in crosstalk noise S1,3 between layout D(Diff) and

layout E(Diff) was very small. It was worth noting that layout E(Diff) had less shielded ground TSV than layout D(Diff), which meant that less ground TSV was used to achieve the same shielding effect, which meant an effectively reduced TSV area. The octagonal TSV array arrangement had great advantages in reducing the area and decreasing the crosstalk noise.

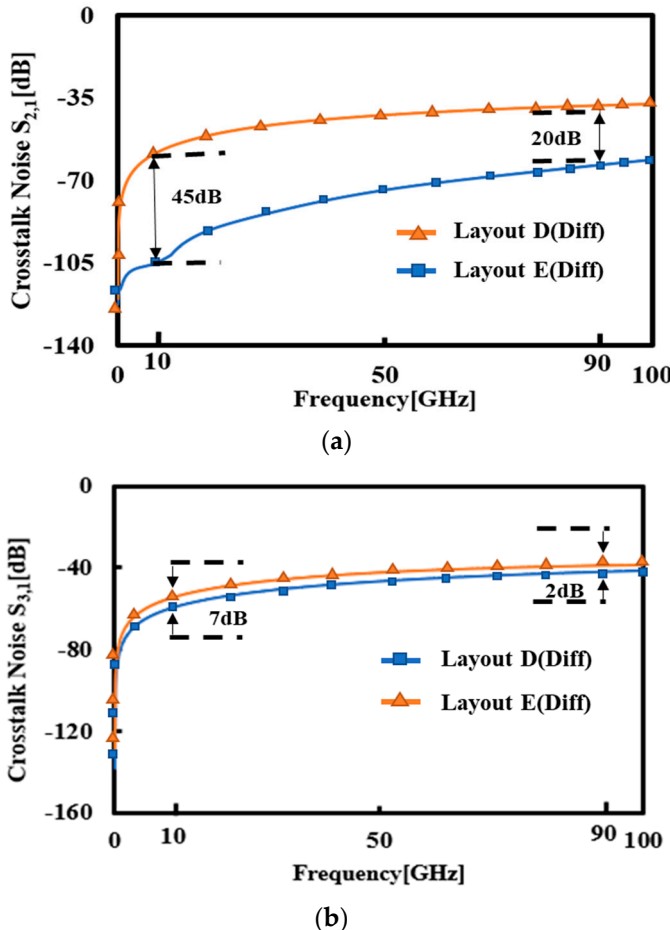

**Figure 11.** The crosstalk noises between layout D(Diff) and layout E(Diff). (**a**) crosstalk noise S2,1 of layout D(Diff) and layout E(Diff); (**b**) crosstalk noise S3,1 of layout D(Diff) and layout E(Diff).

The crosstalk noises among layouts A, B, D(Diff) and E(Diff) are shown in Table 2. Therefore, the layout E has the greatest ability to reduce crosstalk noise in comparison with other arrays.

**Table 2.** The crosstalk noises among layouts A, B, D(Diff) and E(Diff).

| Array | 10 GHz (Crosstalk Noise S2,1) | 10 GHz (Crosstalk Noise S3,1) |
|---|---|---|
| Layout A | −45 dB | - |
| Layout B | −50 dB | - |
| Layout D(Diff) (topology of layout C(Diff)) | −65 dB | −62 dB |
| Layout E(Diff) | −110 dB | −55 dB |

The crosstalk noise between two adjacent TSVs of regular and new structure in only TSV array as shown in Figure 7 was investigated as shown in Figure 12. Compared with regular TSV, the new TSV structure reduced the crosstalk noise by 3 dB at 100 GHz. The main reason may be that the influence path increased between two new TSVs as the

diameter of the new TSV reduced. Then, the regular and new structure applied in TSV arrays as shown in Figures 4b and 8 were simulated by HFSS.

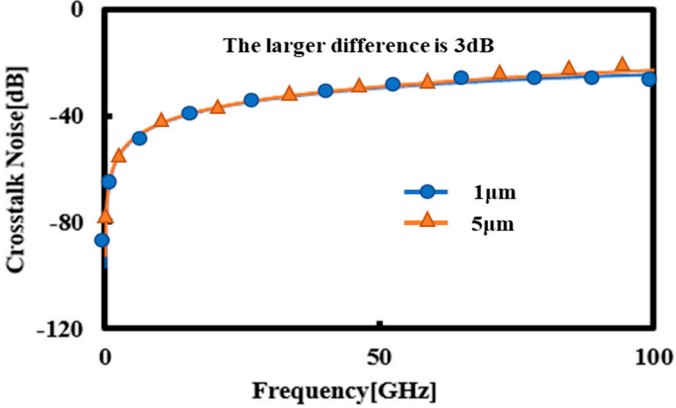

**Figure 12.** The crosstalk noise between two TSVs.

Layout F(Diff) was designed with a smaller TSV being substituted for a large TSV to further decrease the occupied silicon area. The crosstalk noise between layout E(Diff) and F(Diff) is shown in Figure 13. Although the resistance of the new TSV structure was increased by 0.02 Ω, the crosstalk noise did not increase compared with the regular TSV. Notably, the area of a single TSV after optimization was reduced by 7.065 $\mu m^2$ compared to before optimization. The total area of the entire TSV array was reduced by 98.91 $\mu m^2$.

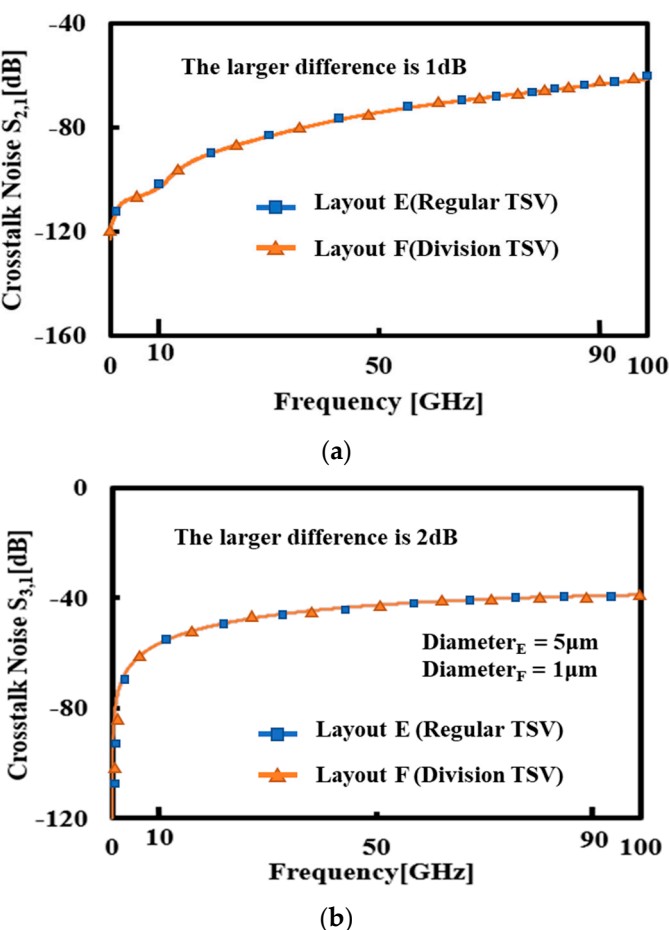

**Figure 13.** The crosstalk noises between layout E and layout F. (**a**) crosstalk noise S2,1 of layout E and layout F; (**b**) crosstalk noise S3,1 of layout E and layout F.

## 4. Conclusions

Compared with the traditional two TSV array layouts (parallel and staggered array) and a hexagonal array, the octagonal TSV array that we proposed had a better ability to suppress the crosstalk noise. When the frequency is over 20 GHz, the crosstalk noise is reduced by 15 dB, 5 dB and 2 dB, respectively.

To further decrease the crosstalk noise, we summarize the effect of two traditional TSV arrays with single-end and differential signals on suppressing crosstalk noise. It is concluded that the staggered layout with differential signals can reduce the coupling by up by 44% at high frequencies in the same area.

On this basis, a differential signal is used in the octagonal TSV array. Compared to the staggered TSV array with a differential signal, the octagonal TSV array with a differential signal can further reduce the crosstalk noise, which can be reduced by about 60% at 10 GHz. At the same time, the number of shielded grounded TSVs is less and the total area of the silicon substrate is smaller.

Furthermore, the design in which one large TSV is changed into four smaller TSVs in the octagonal TSV array can maintain the ability of reducing crosstalk noise in a regular TSV array. Most importantly, the single TSV area is reduced by 7.065 $\mu m^2$ and the total area is reduced by 98.91 $\mu m^2$ when the new TSV structure replaces the regular TSV in an octagonal TSV array.

**Author Contributions:** Conceptualization, Z.L. and H.J.; methodology, H.J.; software, H.J.; Validation, Z.L., H.J. and Z.Z.; formal analysis, H.J.; investigation, H.J.; data curation, H.J.; writing—original draft preparation, H.J.; writing—review and editing, Z.L. and Q.S.; visualization, Z.L. and W.Z.; supervision, Z.Z and L.C.; project administration, Z.L.; funding acquisition Z.L. All authors have read and agreed to the published version of the manuscript.

**Funding:** This work was supported by National Natural Science Foundation of China under Grant 61974056 and 61704056, Natural Science Foundation of Shanghai under Grant 19ZR1471300, and Shanghai Science and Technology Innovation Action Plan under Grant 19511131900.

**Data Availability Statement:** The datasets generated during and/or analysis during the current study are available from the corresponding author on reasonable request.

**Conflicts of Interest:** The authors declare no conflict of interest.

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
