# Peer review of "Crosstalk Noise of Octagonal TSV Array Arrangement Based on Different Input Signal†"

_processes, doi:10.3390/pr10020260_

Round 1

Reviewer 1 Report

1. The introduction does a very poor review of previously published works. The papers are mentioned without highlighting their main contributions or their drawbacks. Therefore, it is not possible to evaluate why this work represents a novel contribution to the state-of-the-art.

2. The work lacks on any technical discussion on the subject. It seems that the authors randomly selected and proposed an octagonal array, but there is no theoretical or heuristic justification on why this could represent a better solution. Why not hexagonal, triangular, or any other shape that differs from the simple parallel array?

3. There is no discussion about the process technology that is being considered in the simulations. Is it a fictional process? 

4. Eight references is a pretty poor revision of previous works. The authors should pursue elaborating a more comprehensive review of the state-of-art.

5. What are the trade-offs introduced by the reduction of the TVS size shown in Figure 4? Is there an increased resistance? Is this size reduction  and new TVS structure compatible with fabrication processes?

6. An extensive English language revision should be done. It is advised a revision performed by a native English speaker.

7. Proof reading is also needed. There are various typos and inconsistencies in the text. 

Reviewer 2 Report

  • Line 45, What is CPW short for. For example, (Through Silicon Via)TSV.
  • Line 130," Fig. 7 obtained by ANSYS HFSS simulation", needs to introduce the simulation in more detail, such as material properties and height of the TSV.
  • Line 132 and Figure 7, "  Compared with array 1 and 2, the crosstalk of array 1 was reduced by 7dB and 2dB when the frequency was over 20GHz." Please clarify where 2dB shows in figure 7 and check the meaning of the sentence (array 1 vs 2, reduce 7dB, array 2 vs 3, reduce 2db)? 
  • Figure 8. what is the definition/schematic drawing of layout C (Diff). is that the same as layout C?
  • Line 172, " Figure 6 (a) showed the crosstalk S21...",  Is that Figure 9(a) instead of Figure 6(a)?
  • Line 224, " four smaller TSVs", please overall check the nouns' plural and singular form.

Author Response

Point 1: Line 45, What is CPW short for. For example, (Through Silicon Via)TSV. 

Response 1: The full name of CPW is coplanar wave guide and the missing part in the article has been supplement.

Point 2: Line 130," Fig. 7 obtained by ANSYS HFSS simulation", needs to introduce the simulation in more detail, such as material properties and height of the TSV.

Response 2: In order to better explore the crosstalk noise among different TSV array, the materials and the height of TSV was set as boundary condition. The materials and height were copper and 50μm, respectively. Meanwhile, the missing part in the article has been supplement.

Point 3: Line 132 and Figure 7, " Compared with array 1 and 2, the crosstalk of array 1 was reduced by 7dB and 2dB when the frequency was over 20GHz." Please clarify where 2dB shows in figure 7 and check the meaning of the sentence (array 1 vs 2, reduce 7dB, array 2 vs 3, reduce 2db).

Response 3: This sentence means that the crosstalk of array 3 can be better reduced by 7dB and 2dB respectively compared with array 1 and 2. The errors in the article have been corrected.

Point 4: Figure 8. what is the definition/schematic drawing of layout C (Diff). is that the same as layout C?

Response 4: The marked as “Diff” means that the input signal was differential signal, which only used in layout C.

Point 5: Line 172, " Figure 6 (a) showed the crosstalk S21...", Is that Figure 9(a) instead of Figure 6(a)? Line 224, " four smaller TSVs", please overall check the nouns' plural and singular form.

Response 5: The figure 6 should be changed to Figure 9 and the singular form should be changed to plural form. I’m so sorry for the difficult review caused by the marking error.

Round 2

Reviewer 1 Report

1. An English language proof-reading is still required.

2. Define acronyms when used for the first time, specially in the Abstract.

3. Include a figure or description of your simulation setup. A screenshot of the Ansys structure would also be useful.

4. Your descriptions of your results can be improved by including a summary comparison in the form of a table, describing the crosstalk for each array structure.

5. This reviewer thinks that the answers provided by the authors in the previous revision to the comments 3 and 5 should be included somehow in the manuscript.

Author Response

Point 1: An English language proof-reading is still required.

Response 1: Thanks for your comment. These errors have been already modified extensively in the manuscript. We present one example as following:

“The influential factors of crosstalk noise between TSVs were investigated including the TSV pitch, signal and ground TSVs location, and signal types (single-end and differential signal) by using coplanar wave guide (CPW) testing structure. These results based on traditional TSV arrays showed that staggered TSV layout with differential signals had lower crosstalk noise”

Point 2: Define acronyms when used for the first time, especially in the Abstract.

Response 2: I highly appreciate your comment. The errors in the abstract have been modified and others have been checked carefully. Following is the example :

“This paper proposed an octagonal layout for enhancing the ability of resisting electromagnetic interference in Through Silicon Via (TSV) array. The influential factors of crosstalk noise between TSVs were investigated including the TSV pitch, signal and ground TSVs location, and signal types (single-end and differential signal) by using coplanar wave guide (CPW) testing structure.”

Point 3: Include a figure or description of your simulation setup. A screenshot of the Ansys structure would also be useful.

Response 3: Thank you so much for your comment. In order to better explain the results of this work, the simulation model in Ansys HFSS already supplement as below:

  “However, further decreasing the crosstalk needs more researches, especially simultaneously reducing the area occupied by TSVs. Therefore, this paper proposed an octagonal TSV array arrangement based on the studies referred above and the CPW structure was used to measure the crosstalk as Fig. 2 showed.

Point 4: Your descriptions of your results can be improved by including a summary comparison in the form of a table, describing the crosstalk for each array structure.

Response 4: I really appreciate your comment. The results have been summarized in a table.

“The crosstalk among layout A, B, D(Diff) and E(Diff) were shown in table 2. Therefore, the layout E had the greatest ability to reduce crosstalk by comparing with other arrays.

Point 5: This reviewer thinks that the answers provided by the authors in the previous revision to the comments 3 and 5 should be included somehow in the manuscript.

Response 5: Thanks for your comment. In fact, the comments 3 and 5 in the previous revision have been added to the manuscript but did not highlight in the previous manuscript.

This version was all highlighted as below:

 “In the existing process, the diameter of TSV can be manufactured as small as 100nm by using a Bosch process applied to extremely small CD structures (180x250nm top CD) [12], thus the new TSV structure could be manufactured.”

“However, the research of reducing crosstalk noise in TSV array was seldom found and more research focused on the process and thermal stress of TSV in recent year. Meanwhile, there was little current research on using differential signals in TSV array like references [10-11].”

“Although the resistance of new TSV structure was increased by 0.02 Ω, the crosstalk noise didn’t increase compared with regular TSV.”

Round 3

Reviewer 1 Report

There are no further comments.

Author Response

Thanks for your review.

This manuscript is a resubmission of an earlier submission. The following is a list of the peer review reports and author responses from that submission.

Round 1

Reviewer 1 Report

The authors propose the simulation of an octagonal layout of TSVs that reduces the crosstalk between signals in a 3D integrated circuit design.

The work has already been published practically in its entirety by the same authors in an IEEE conference (reference 2).

The proposal made is the same as in the previous work, the results, graphs and conclusions derived from the discussion as well. Only the simulation of single-ended signals in an octagonal layout is included as a novelty.

I believe that the article does present nothing new to justify another publication of the work in this journal.

The authors should provide new proposals, new studies and experiments, and not just a new simulation so that the work can be considered for publication.

Reviewer 2 Report

The authors are encouraged to consider the following comments:

1. Please, define all your acronyms when used for the first time in the manuscript.

2. The introduction does a very poor review of previously published works on the subject. The papers are mentioned without highlighting their main contributions or their drawbacks. Therefore, it is not possible to evaluate why this work at hand represents a novel contribution to the state-of-the-art.

3. The work lacks on any technical discussion on the subject. It seems that the authors randomly selected and proposed an octagonal array, but there is no theoretical justification on why this could represent a better arrangement that traditional ones. Why not hexagonal, triangular, or any other shape that differs from the simple parallel array?

4. There is no discussion to what process technology is being considered in the simulations. Is it a fictional process?

5. Eight references is a pretty poor revision of previous works. The authors should pursue elaborating a more comprehensive review of the state-of-art.

6. The manuscript in general is very short. It should be submitted as a brief and not as a research article.

7. An extensive English language revision should be done. It is advised a revision performed by a native English speaker.